# ON THE ROLE OF PRE-TRAINING FOR META FEW-SHOT LEARNING

## ABSTRACT

Few-shot learning aims to classify unknown classes of examples with a few new examples per class. There are two key routes for few-shot learning. One is to (pre-)train a classifier with examples from known classes, and then transfer the pre-trained classifier to unknown classes using the new examples. The other, called meta few-shot learning, is to couple pre-training with episodic training, which contains episodes of few-shot learning tasks simulated from the known classes. Pre-training is known to play a crucial role for the transfer route, but the role of pre-training for the episodic route is less clear. In this work, we study the role of pre-training for the episodic route. We find that pre-training serves as major role of disentangling representations of known classes, which makes the resulting learning tasks easier for episodic training. The finding allows us to shift the huge simulation burden of episodic training to a simpler pre-training stage. We justify such a benefit of shift by designing a new disentanglement-based pre-training model, which helps episodic training achieve competitive performance more efficiently.

## 1 INTRODUCTION

In recent years, deep learning methods have outperformed most of the traditional methods in supervised learning, especially in image classification. However, deep learning methods generally require lots of labeled data to achieve decent performance. Some applications, however, do not have the luxury to obtain lots of labeled data. For instance, for bird classification, an ornithologist typically can only obtain a few pictures per bird species to update the classifier. Such needs of building classifiers from limited labeled data inspire some different research problems, including the few-shot learning problem (Finn et al., 2017; Snell et al., 2017; Rajeswaran et al., 2019; Oreshkin et al., 2018; Vinyals et al., 2016; Lee et al., 2019). In particular, few-shot learning starts with a training dataset that consists of data points for "seen" classes, and is required to classify "unseen" ones in the testing phase accurately based on limited labeled data points from unseen classes.

Currently, there are two main frameworks, meta-learning (Finn et al., 2017; Snell et al., 2017; Chen et al., 2019) and transfer learning (Dhillon et al., 2020), that deal with the few-shot learning problem. For transfer learning, the main idea is to train a traditional classifier on the meta-train dataset. In the testing phase, these methods finetune the model on the limited datapoints for the labeled novel classes. For meta-learning frameworks, their main concept is episodic training (Vinyals et al., 2016). For the testing phase of few-shot learning, the learning method is given $N$ novel classes, each containing $K$ labeled data for fine-tuning and $Q$ query data for evaluation. Unlike transfer learning algorithms, episodic training tries to simulate the testing literature in the training phase by sampling episodes in training dataset.

In these two years, some transfer-learning methods (Dhillon et al., 2020) with sophisticated design in the finetuning part have a competitive performance to the meta-learning approaches. Moreover, researchers (Lee et al., 2019; Sun et al., 2019; Chen et al., 2019; Oreshkin et al., 2018) have pointed out that combining both the global classifier (pre-training part) in the transfer learning framework and the episodic training concept for the meta-learning framework could lead to better performance. Yet, currently most of the attentions are on the episodic training part (Vinyals et al., 2016; Finn et al., 2017; Snell et al., 2017; Oreshkin et al., 2018; Sun et al., 2019; Lee et al., 2019) and the role of pre-training is still vague.

Meta-learning and pre-training has both improved a lot in the past few years. However, most of the works focus on the accuracy instead of the efficiency. For meta-learning, to make the progress more efficient, an intuitive way is to reduce the number of episodes. Currently, there are only limited researches (Sun et al., 2019) working on reducing the number of episodes. One of the methods (Chen et al., 2019; Lee et al., 2019) is to apply a better weight initialization method, the one from pre-training, instead of the random initialization. Another method (Sun et al., 2019) is to mimic how people learn. For example, when we are learning dynamic programming, given a knapsack problem with simple constraint and the one with strong constraint, we will learn much more when we solve the problem with strong constraint. Sun et al. (2019) followed the latter idea and crafted the hard episode to decrease amount of necessary episodes.

In this work, we study the role of pre-training in meta few-shot learning. We study the pre-training from the disentanglement of the representations. Disentanglement is the property that whether the datapoints within different classes has been mixed together. Frosst et al. (2019) pointed out that instead of the last layer of the model all representations after other layers were entangled. The last layer does the classifier and the rest captures some globally shared information. By analyzing the disentanglement property of episodic training, though the pre-training gives a better representation that benefits the episodic training, the representation becomes more disentangled after episodic training. That is to say, episodic training has spent some effort on making the representation more disentangled. Benefited from the understanding, we design a sophisticated pre-training method that is more disentangled and helps episodic training achieve competitive performance more efficiently. With our pre-training loss, the classical meta-learning algorithm, ProtoNet (Snell et al., 2017), achieves competitive performance to other methods. Our study not only benefits the episodic training but also points out another direction to sharpen and speed-up episodic training.

To sum up, there are three main contributions in this work:

1. A brief study of the role of pre-training in episodic training.
2. A simple regularization loss that sharpens the classical meta-learning algorithms.
3. A new aspect for reducing the necessary episodic training episodes.

## 2 Related Work

Few-shot learning tries to mimic the human ability to generalize to novel classes with limited datapoints. In the following, we briefly introduce the recent progress of the transfer-learning framework and two categories of the meta-learning framework. Afterward, we give a brief introduction of the not well studied episode efficiency problem.

### 2.1 Transfer-Learning Framework

In the training phase, the transfer-learning framework trains a classifier on the general classification task across all base classes instead of utilizing episodic training. And for the testing phase, transfer-learning methods finetune the model with the limited labeled data. There are several kinds of tricks. Qi et al. (2018) proposed a method to append the mean of the embedding with a given class as a final layer of the classifier. Qiao et al. (2018) used the parameter of the last activation output to predict the classifier for novel classes dynamically. Gidaris & Komodakis (2018) proposed a similar concept with Qiao et al. (2018). They also embedded the weight of base classes during the novel class prediction. Moreover, they introduced an attention mechanism instead of directly averaging among the parameters of each shot. Besides embedding base classes weight to the final classifier, Dhillon et al. (2020) utilized label propagation by the uncertainty on a single prediction to prevent overfitting in the finetune stage, which is quite similar to the classical classification tasks.

### 2.2 Meta-Learning Framework

For meta-learning like framework, the main concepts are learning to learn and episodic training (Vinyals et al., 2016). Learning to learn refers to learn from a lot of tasks to benefit the new task learning. To prevent confusion, the original train and test phase are regarded as "meta-train" and "meta-test". The term "train" and "test" would be referred to the one in each small task. Episodic

training is the process of mimicking the task structure in meta-test during training. If the meta-test phase consists of $K$ support examples and $Q$ query examples from $N$ classes, then we will sample lots of tasks that also have $K$ support examples and $Q$ query examples from $N$ classes in the meta-train phase. Meta-learning algorithms have developed rapidly in recent years. We briefly categorize them into two categories, optimization-based methods, metric-based methods.

**Optimization-based Method**  Optimization-based methods try to get an embedding that could easily fit subtasks by adding some extra layers. Finn et al. (2017) proposed MAML (Model-Agnostic Meta-Learning), which got a network that was the closest to all the best model in the low-way low-shot tasks. However, MAML might be quite inefficient due to the computation of Hessian matrix. To leverage the issue, iMAML (Rajeswaran et al., 2019) and foMAML (Finn et al., 2017) provides different approximation to avoid the heavy computation. However, MAML still suffers from the high dimensional overfitting issue. LEO model (Rusu et al., 2019) solves the overfitting issue by learning a low dimensional latent space. Instead of aiming to get an embedding that could benefit the latter fully connected layer, MetaOptNet (Lee et al., 2019) aims to get an embedding that could benefit the latter differentiable support vector machine.

**Metric-based Method**  Instead of learning an embedding that could benefit the latter additional layer, metric-based methods aim to get an embedding that could easily classify the classes by simple metrics. Matching Networks (Vinyals et al., 2016) conducts a cosine similarity metric with a Full Context Encoding module. Prototypical Networks (Snell et al., 2017) replaces the cosine similarity metric with the squared Euclidean distance and computes the mean of the embedding in the support-set as the prototype. Relation Network (Sung et al., 2018) embeds a relation module in the learning metric. Instead of using a consistent metric in the task, TADAM (Oreshkin et al., 2018) designs a task-dependent metric that could dynamically fit new class combinations.

## 2.3 MIXED FRAMEWORK

Some recent works have found that using a global pre-trained classifier as the initialization weight could lead to a better meta-learning result. Sun et al. (2019) used the pre-trained classifier weight as initialization weight and launched a simple gradient-based with restricting the learning process as shift and scale. Meta-Baseline (Chen et al., 2020) also follows the initialization literature and applies cosine similarity metric for the following learning process. Chen et al. (2019) changed the original pre-trained network structure into a weight imprinting taste and a simple gradient-based method for the episodic training part. Triantafillou et al. (2020) also utilized the pre-trained initialization and derived a combination between MAML Finn et al. (2017) and ProtoNet (Snell et al., 2017).

## 2.4 EPISODE REDUCTION METHODS

Recent researchers have found that a pre-trained classifier leads to better meta-learning results. On the other hand, we could reduce the amount of episodes by using a pre-trained classifier. Besides utilizing the pre-training weight to reduce the number of episodes, Meta Transfer Learning (Sun et al., 2019) proposes the concept of hard episode. For each normal episode, MTL adds the class with the worst performance in a pool. After collecting for a while, MTL creates hard episodes by sampling from the pool.

Instead of crafting hard episodes, our approach tries to utilize more in the pre-training phase. We propose a simple regularization that could reduce the difference between the embeddings of the pre-trained classifier and the episodic training one. It has significantly reduced the number of episodes and achieves a similar (even better) performance for the original algorithms. Moreover, for shallow and deep backbones, it increases the final accuracy.

## 3 METHODOLOGY

No matter in the route of transferring the classifier or the route of episodic training, pre-training serves a crucial role. And in meta few-shot learning, pre-training provides an initialization weight for further episodic training. In recent episodic-based methods, the pre-training model is split into two parts, backbone and classifier. The last linear layer serves as the classifier and maps the embedding

to logits. Others work as the backbone and transform the raw photo into embedding. After pre-training, the classifier part is directly dropped, since the target classes may be unseen or the order may have changed. Though the split is quite naive, the afterward episodic learning converges faster and better based on the initialization. Thus, previous works conclude that pre-training provides a *better* representation. However, what makes it *better* is not clear. What is the role of pre-training in meta few-shot learning? More specifically, what is the character of backbone in pre-training and episodic training?

In ths section, we choose prototypical network (Snell et al., 2017) as the representative. Following the analysis of general learning literature by Frosst et al. (2019), we utilize the similar loss to measure the disentanglement and entanglement property of the backbone in pre-training and episodic training. Benefited from our observation, we give an attempt to transfer the computation burden of episodic training by adding an sophisticated loss in pre-training.

## 3.1 NOTATION

In a $N$-way $K$-shot few-shot classification task, we are given a small support set of $NK$ labeled data $\mathbb{S} = \{(x_1, y_1), ..., (x_{NK}, y_{NK})\}$ where $x_i \in \mathbb{R}^D$ and $y_i \in \{1, ..., N\}$. $\mathbb{S}_n$ denotes the set of examples labeled with class $n$.

## 3.2 PROTOTYPICAL NETWORK

Prototypical Network (Snell et al., 2017) is one of the classical metric-based meta few-shot learning algorithms. First, it computes prototypes $c_i$, which are $M$-dimensional representations $c_i \in \mathbb{R}^M$ by averaging the output of the embedding function $f_\theta : \mathbb{R}^D \to \mathbb{R}^M$ from the same class set $\mathbb{S}_i$.

$$c_i = \frac{1}{|\mathbb{S}_i|} \sum_{(\boldsymbol{x}, \boldsymbol{y}) \in \mathbb{S}_i} f_\theta(\boldsymbol{x}) \tag{1}$$

Then, the prototypical network calculates a probability distribution for a given data $x$ by softmax over the Euclidean distance $d$ between each prototype and the embedding $f_\theta(x)$.

$$p_\theta(y = n | \boldsymbol{x}) = \frac{exp(-d(f_\theta(x), \boldsymbol{c}_n))}{\sum\limits_{n' \in \{1, ..., N\}} exp(-d(f_\theta(x), \boldsymbol{c}_{n'}))} \tag{2}$$

## 3.3 SOFT-NEAREST NEIGHBOR LOSS

"Soft-Nearest Neighbour loss [...] is proposed by Frosst et al. (2019)" Frosst et al. credits Salakhutdinov and Hinton (2007) for the soft nearest neighbour loss, which itself draws inspiration from Goldberger et al. (2005)'s Neighbourhood Component Analysis.

Soft-Nearest Neighbor Loss (SNN-Loss, Eq.3) is first proposed by Salakhutdinov & Hinton (2007). Then, Frosst et al. (2019) applies the loss to analyze disentanglement of layer. The loss measures the disentanglement property of a representation. Higher loss means more entangled representation. In the original study, general supervised learning, which is the pre-training part in meta few-shot learning, has an entanglement trend in all the layers instead of the last layer. On the other hand, in meta few-shot learning, especially metric-based methods, after the pre-training phase, the last layer is replaced with a metric. This leads to an interesting question. *What should the afterward last layer be, entangled or disentangled?* By the experiment in Sec.4.4, we find that with sufficient amount of pre-training the representation is less entangled after episodic learning. In other words, the metric prefers a more disentangled representation.

$$\textbf{Soft-Nearest-Neighbor-Loss} := -\frac{1}{b} \sum_{i \in \{1..b\}} \frac{\sum\limits_{j \in \{1..b\}, j \neq i, y_i = y_j} exp(-\|x_i - x_j\|^2)}{\sum\limits_{n \in \{1..b\}, n \neq i} exp(-\|x_i - x_n\|^2)} \tag{3}$$

---

**Algorithm 1** Regularized Pre-trained Prototypical Network (RP-Proto)

---

**Require:** Training set $\mathbb{D} = \{(\boldsymbol{x}_{11}, \boldsymbol{y}_{11}), ...., (\boldsymbol{x}_{NK}, \boldsymbol{y}_{NK})\}$, where each $\boldsymbol{y}_{nk} = n$.
 1: $M$: metric for judging distance between two vector.
 2: $f_\theta$: stands for the feature extractor which may output $e$ dimension vector.
 3: $g$: maps the output of $f_\theta$ to $k$ dimension.
 4: $b$: batch sizes for pretraining.
 5: $p$: pretrained iteration number.
 6: $\alpha$: weight vector for the regularization loss.
 7: $L$: cross entropy loss
 8: Initialize $W_i$ with $e$ dimension for $i \in \{1, ..., K\}$
 9: **for** $i \in range(p)$ **do**
10:     $(X, Y) \leftarrow \text{sample}(\mathbb{D}, b)$;
11:     $l_c \leftarrow L(g(f_\theta(X)), Y)$
12:     $l_{reg} \leftarrow \frac{1}{b} \sum\limits_{(\boldsymbol{x},\boldsymbol{y}) \in (\mathbb{X},\mathbb{Y})} M(f_\theta(x), W_y)$
13:     $l_{total} \leftarrow l_c + \alpha \times l_r$
14:     $f_\theta, g \leftarrow \text{backprop}(l_{total}, (f_\theta, g))$
15: **end for**
16: $f_\theta \leftarrow \text{ProtoNet}(f_\theta, D)$
17: **return** $f_\theta$

---

## 3.4 Regularized Pre-trained Prototypical Network (RP-Proto)

In the previous section, we conclude that the last layer in the backbone would be more disentangled after episodic training. As a result, we wonder whether providing a more disentangled representation of the penultimate layer in the pre-training part could speed up the later episodic training. A naive way to provide a more disentangled representation may be transferring the loss in the episodic training part to pre-training. However, the naive method slows down the pre-training part due to the episode construction. Also, it may require an additional sampler to craft the episode task. To prevent from additional sampler, we could directly compute the episodic training loss among each batch during pre-training. However, the naive method suffers from three critical issues. First, if the batch size is not in the scale of the number of pre-training classes, some class may not exist in the batch. Second, the amount of each class may be non-uniform. Third, the computation of the gradient of a single instance may mix up with other instance in the batch, which makes parallelization harder.

To deal with the above issues, we introduce a surrogate loss, which aims for *an embedding with more disentanglement*. We capture the disentanglement representation from another aspect, *instances within the same class should gather together in the embedding space*. The surrogate loss $l_{reg}$ is based on the distance between $f_\theta(x)$ and $W_y$, where $W_y$ is an $M$-dimensional learnable parameter.

$$l_{reg}(x, y) = d(f_\theta(x), W_y) \tag{4}$$

For the pre-training phase on the ordinary classification task, $f_\theta$ corresponds to the network right before mapping to a final $K$-dimension embedding. Then, we add $l_{reg}$ to the ordinary classification loss with a multiplying weight of $\alpha$.

$$l_{total} = l_{classification}(\boldsymbol{x}, \boldsymbol{y}) + \alpha \times l_{reg}(\boldsymbol{x}, \boldsymbol{y}) \qquad (\alpha = 10^C) \tag{5}$$

$W_y$ is the surrogate mean of each class. Instead of directly summing up all the embeddings, $W_y$ could be calculated by backpropagation. Moreover, when $W_y$ equals to zero, it could be considered as an $L2$ regularization on the feature space, which scales the embedding. When $W_y$ is learnable, it makes each instance in the same class closer which satisfies the our goal of *disentangled representation*. We have described the detail in Algo.1.

Regarding the flaws of the naive approach, in this auxiliary loss schema, a data point doesn't need to consider the distance of data points in other classes, As a result, it could avoid the large computation effort of the cross-classes datapoints distance and the non-uniform amounts of datapoints issue.

## 4 EXPERIMENTS

### 4.1 DATASET DETAILS

**MiniImagenet**   Proposed by Vinyals et al. (2016) is a widely used benchmark for few-shot learning. It is a subset of ILSVRC-2015 (Deng et al., 2009) with 100 classes, 64 for meta-train, 16 for meta-validation and 20 for meta-test. Each class contains 600 images of size 84x84.

**CIFAR-FS & FC100**   Proposed by Bertinetto et al. (2019) and Oreshkin et al. (2018), both are splits between the original classes of CIFAR100 (Krizhevsky et al.). They also follow the mini-Imagenet structure with 64 base classes, 16 validation classes, and 20 novel classes. The pictures in CIFAR100 are 32 x 32 low resolution. The main difference is that CIFAR-FS randomly splits among the classes, but FC100 splits the classes carefully with less semantic overlap between the base classes and the novel classes. In general, FC100 is more difficult than CIFAR-FS.

### 4.2 EXPERIMENT SETUP

First, we design an experiment to study the role pre-training in meta-fewshot learning. We train several classifiers with sufficient epochs and launch the episodic training with the prototypical network methods. The disentanglement of the penultimate layer in pre-training and the last layer in episodic training are compared. The experiment is evaluated in 5-way 5-shot settings.

Second, to evaluate the power of the auxiliary loss and validate our conjecture. We compare the episodic training performance based on the backbones with and without our auxiliary loss in pre-training phase. The experiment is evaluated in both 5-way 5-shot and 5-way 1-shot settings. In more detail, we search the auxiliary loss weighted coefficient $\alpha = 10^C$ from $C = -1$ to $-3.5$ for each backbone on each dataset.

### 4.3 IMPLEMENTATION DETAILS

**Model Architecture**   For the backbone of ResNet10 and ResNet18, we follow the preprocessing from Chen et al. (2019). We use image sizes with 224x224 for both pre-trained prototypical network and regularization pre-trained prototypical network. For ResNet12 backbone, we follow the implementation from Sun et al. (2019) and use image sizes with 80x80. For the Conv4 backbone, we follow the implementation in prototypical network (Snell et al., 2017) with image size 84x84. The detail parameters are in the appendix.

**Pre-training**   For pre-training the backbone, we follow the settings in Chen et al. (2020). We use a stochastic gradient descent optimizer with a learning rate as 0.001 for all the backbones and the batch size is 128.

**Episodic training**   For the episodic training part, we also follow the settings in Chen et al. (2020). We use stochastic gradient descent optimizer with learning rate as 0.001 and turn off the data augmentation part in the preprocess pipeline to reach a more stable result.

**Evaluation**   For the performance evaluation of episodic training, we turn off all the data augmentation and record the mean of the performance and the 95% confidence intervals among 600 randomly sampled episodes.

### 4.4 LAYER DISENTANGLEMENT WITH METRIC

In this part, we design experiments to figure out the role of pre-training. We utilize the soft-nearest neighbor loss Eq.3 to measure the disentanglement of the last layer in episodic training among episodes. Fig.1 is the soft-nearest loss in episodic training. No matter in the shallow backbone Conv4 or the deep backbone ResNet10, ResNet18 the loss has a decreased trend while the episode increases. For lower soft-nearest-neighbor loss the representation becomes more disentangled. That is to say, in episodic learning, the metric prefers a less entangled representation. This makes sense in the higher level. Since in episodic training, especially metric-based methods, the metric may

Table 1: 5-way 5-shot and 5-way 1-shot performance on miniImagenet. The weighted coefficient of the regularization loss is 10 to C.

|  | BACKBONE | 1 SHOT | 5 SHOT |
|---|---|---|---|
| Matching Networks | Conv4 | $43.56 \pm 0.84$ | $55.31 \pm 0.73$ |
| Prototypical Networks | Conv4 | $48.70 \pm 1.84$ | $63.11 \pm 0.92$ |
| LEO | WRN-28-10 | $61.76 \pm 0.08$ | $77.59 \pm 0.12$ |
| Chen et al. | ResNet18 | $51.87 \pm 0.77$ | $75.68 \pm 0.63$ |
| SNAIL | ResNet12 | $55.71 \pm 0.99$ | $68.88 \pm 0.92$ |
| AdaResNet | ResNet12 | $56.88 \pm 0.62$ | $71.94 \pm 0.57$ |
| TADAM | ResNet12 | $58.50 \pm 0.30$ | $76.70 \pm 0.30$ |
| MTL | ResNet12 | $61.2 \pm 1.80$ | $75.50 \pm 0.80$ |
| MetaOptNet | ResNet12 | $62.64 \pm 0.61$ | $78.63 \pm 0.46$ |
| Prototypical Networks (Pre) | Conv4 | $42.08 \pm 0.75$ | $67.77 \pm 0.68$ |
| Prototypical Networks (Pre) | ResNet10 | $55.34 \pm 0.81$ | $76.97 \pm 0.67$ |
| Prototypical Networks (Pre) | ResNet12 | $56.83 \pm 0.83$ | $76.45 \pm 0.64$ |
| Prototypical Networks (Pre) | ResNet18 | $57.80 \pm 0.84$ | $77.71 \pm 0.63$ |
| RP-Proto (euclidean) (C: -1.5) | Conv4 | $50.38 \pm 0.80$ | $69.56 \pm 0.71$ |
| RP-Proto (euclidean) (C: -1.5) | ResNet10 | $56.92 \pm 0.84$ | $77.45 \pm 0.65$ |
| RP-Proto (euclidean) (1-shot C: -2.5) (5-shot C: -3.5) | ResNet12 | $57.84 \pm 0.82$ | $77.08 \pm 0.70$ |
| RP-Proto (euclidean) (1-shot C: -1.0) (5-shot C: -3.5) | ResNet18 | $57.88 \pm 0.86$ | $77.43 \pm 0.63$ |

be geometric related, e.g. the distance to the center of each class. If the representation is highly entangled, it is non-intuitive for the metric to figure the class of each instance.

## 4.5 PERFORMANCE EVALUATION

In the miniImagenet dataset result (Table.1), our method shows competitive performance to other methods. We find that adding the auxiliary loss could make the shallow network has a similar performance with the large one. Moreover, when the backbone is shallow, we have outperformed all other methods with the same backbone. For experiments on fc100 and cifar-fs, the results are similar to miniImagenet. For detailed results, please check the appendix.

For our major conjecture, *disentangled representations could benefit the following episodic training*, we do a one

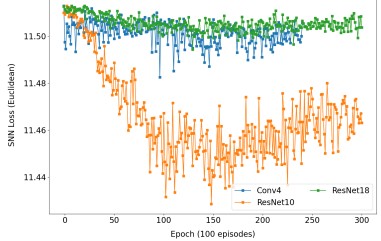

Figure 1: Comparison for the SNN-Loss during the meta-learning process on 5-way 5-shot miniImagenet.

by one comparison between general pre-trained Prototypical Networks and the regularized pre-trained one by recording the meta-test performance after every 100 episodes in Fig.2. Instead of ResNet18, all the regularized pre-trained backbones lead to a faster convergent speed by providing better initial accuracy. This could support our conjecture that disentangled representations could boost the speed of episodic training.

## 4.6 REGULARIZATION AND DISENTANGLEMENT

Our goal of the regularization loss is to control the disentanglement of the embedding space. In the previous experiments, our methods indeed benefit the speed of episodic training. However, how the regularization affects the disentanglement and the performance is still not clear. There are two potential reasons for interaction. First, the regularized pre-training leads the model to start in a better initialization point with higher disentanglement and higher accuracy. Second, the regularized term helps the model to start in an initialization space with better gradient property to achieve higher disentanglement and optimal weight.

Fig.3 shows the experiment in 5-way 5-shot miniImagenet. The SNN-loss with regularization is always lower than the original one. However, it doesn't support the first possible reason, since the

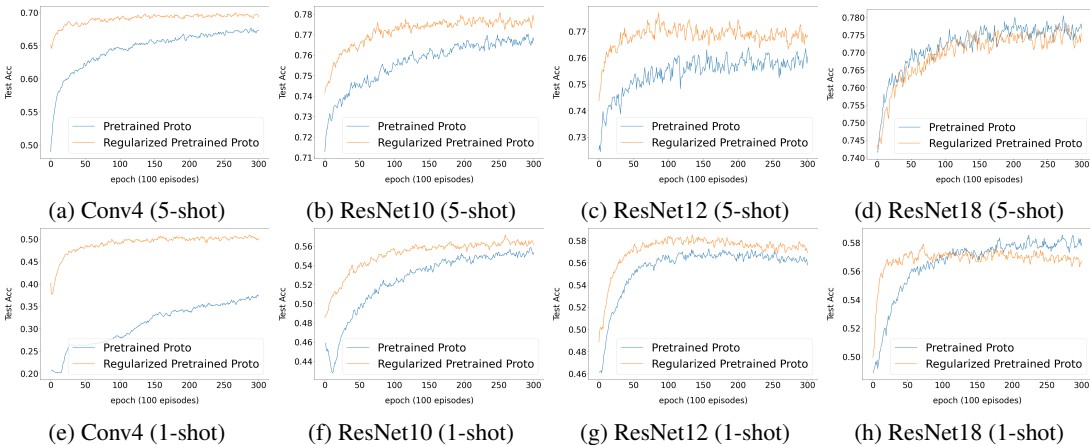

(a) Conv4 (5-shot)  (b) ResNet10 (5-shot)  (c) ResNet12 (5-shot)  (d) ResNet18 (5-shot)

(e) Conv4 (1-shot)  (f) ResNet10 (1-shot)  (g) ResNet12 (1-shot)  (h) ResNet18 (1-shot)

Figure 2: Comparison for the converge speed between standard pre-trained and regularized pre-trained on 5-way 5-shot and 5-way 1-shot miniImagenet.

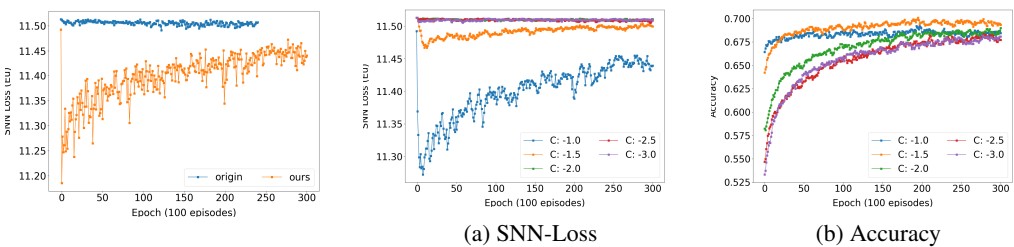

(a) SNN-Loss  (b) Accuracy

Figure 3: SNN-loss ablation comparison with regularization.

Figure 4: Accuracy and SNN-loss with different weighted parameter in episodic training

initial SNN-loss is quite similar to the one without regularization. However, after several episodes, the loss severely decreases. This makes the second claim more convincing.

In Fig.4a, larger weighted parameter leads to higher disentanglement. And in Fig.4b, there is a large gap for the initialization accuracy for different weighted parameters. However, better initialization accuracy doesn't imply a better final accuracy. For instance, $C = -1.5$ has a worse initialization accuracy but converges to the best weight in the end. With suitable weighted parameters, the representation is disentangled properly, the final accuracy has a great improvement. It is similar to the general regularization idea that suitable regularization could lead to performance improvement but extreme regularization may harm the performance.

## 5 CONCLUSION

The disentanglement analysis provides a new aspect to view the role of pre-training in meta few-shot learning. Furthermore, the relationship between the backbone and the classifier is discussed. Though in higher concept, the metric should replace the role of pre-training classifier in the episodic training phase. However, the backbone needs to share the role of the classifier, so the backbone has a disentangled trend in the classifier. Benefited from the observation, we designed an auxiliary loss to transfer the computation burden of episodic training to the pre-training. With the help of the loss, episodic training could achieve competitive performance with fewer episodes.

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

## A APPENDIX

### A.1 PREPROCESS PIPELINE

Table 2: Preprocess pipeline

| FUNCTION | PARAMETER |
| --- | --- |
| RandomSizedCrop | image_size |
| RandomHorizontalFlip | |
| Normalize Mean | (0.485, 0.456, 0.406) |
| Normalize Std | (0.229, 0.224, 0.225) |

### A.2 PEFORMANCE EVALUATION ON FC100 AND CIFAR-FS

Table 3: 5-way 5-shot and 5-way 1-shot performance on CIFAR-FS.

| | BACKBONE | 1 SHOT | 5 SHOT |
|---|---|---|---|
| MAML | Conv4 | $58.9 \pm 1.9$ | $71.5 \pm 1.0$ |
| Prototypical Networks | Conv4 | $55.5 \pm 0.7$ | $72.0 \pm 0.6$ |
| Relation Networks | 64-96-128-256 | $55.0 \pm 1.0$ | $69.3 \pm 0.8$ |
| R2D2 | 96-192-384-512 | $65.3 \pm 0.2$ | $79.4 \pm 0.1$ |
| MetaOptNet | ResNet12 | $72.8 \pm 0.7$ | $85.0 \pm 0.5$ |
| Prototypical Networks (Pre) | Conv4 | $39.94 \pm 0.81$ | $74.19 \pm 0.73$ |
| Prototypical Networks (Pre) | ResNet10 | $63.45 \pm 0.93$ | $81.62 \pm 0.65$ |
| Prototypical Networks (Pre) | ResNet12 | $67.58 \pm 0.92$ | $83.25 \pm 0.64$ |
| Prototypical Networks (Pre) | ResNet18 | $65.65 \pm 0.92$ | $82.80 \pm 0.62$ |
| RP-Proto (euclidean) (C: -1) | Conv4 | $54.66 \pm 0.89$ | $75.16 \pm 0.75$ |
| RP-Proto (euclidean) (C: -1) | ResNet10 | $66.20 \pm 0.93$ | $83.77 \pm 0.69$ |
| RP-Proto (euclidean) (C: -3) | ResNet12 | $67.52 \pm 0.93$ | $83.30 \pm 0.63$ |
| RP-Proto (euclidean) (C: -3) | ResNet18 | $65.34 \pm 0.90$ | $82.98 \pm 0.68$ |

Table 4: 5-way 5-shot and 5-way 1-shot performance on fc100.

| | BACKBONE | 1 SHOT | 5 SHOT |
|---|---|---|---|
| Prototypical Networks | Conv4 | $35.3 \pm 0.6$ | $48.6 \pm 0.6$ |
| TADAM | ResNet12 | $40.1 \pm 0.4$ | $56.1 \pm 0.4$ |
| MetaOptNet | ResNet12 | $47.2 \pm 0.6$ | $62.5 \pm 0.6$ |
| Prototypical Networks (Pre) | Conv4 | $35.63 \pm 0.72$ | $51.80 \pm 0.73$ |
| Prototypical Networks (Pre) | ResNet10 | $38.55 \pm 0.70$ | $57.71 \pm 0.71$ |
| Prototypical Networks (Pre) | ResNet12 | $39.48 \pm 0.66$ | $57.18 \pm 0.72$ |
| Prototypical Networks (Pre) | ResNet18 | $39.02 \pm 0.69$ | $56.57 \pm 0.73$ |
| RP-Proto (euclidean) (C: -1.5) | Conv4 | $37.36 \pm 0.68$ | $53.91 \pm 0.70$ |
| RP-Proto (euclidean) (1-shot C: -1) (5-shot C: -2) | ResNet10 | $39.03 \pm 0.69$ | $57.96 \pm 0.71$ |
| RP-Proto (euclidean) (C: -3) | ResNet12 | $39.73 \pm 0.69$ | $56.11 \pm 0.72$ |
| RP-Proto (euclidean) (C: -3) | ResNet18 | $38.88 \pm 0.67$ | $56.93 \pm 0.73$ |

