# OpenReview forum: "On the Role of Pre-training for Meta Few-Shot Learning"
_ICLR.cc/2021/Conference — Reject_

### Official Review · AnonReviewer1 · 2020-10-15
**Studies an interesting problem, but lacks an important comparison with previous work and is experimentally weak.**

**Rating:** 3
**Confidence:** 4

**Review:**

Summary
========
This paper investigates the role of pre-training as an initialization for meta-learning for few-shot classification. In particular, they look at the extent to which the pre-trained representations are disentangled with respect to the class labels. They hypothesize that this disentanglement property of those representations is responsible for their utility as the starting point for meta-learning. Motivated by this, they design a regularizer to be used during the pre-training phase to encourage this disentanglement to be even more prominent with the hope that this pre-trained solution is now closer to the optimal one, thus requiring less additional episodic training which is time-consuming. They show experimentally that their modified pre-training phase sometimes leads to better results as an initialization for Prototypical Networks compared to the standard pre-trained solution, and sometimes converges faster.

Pros
====
The topic of study of this paper is very interesting. I definitely agree that the role of each of the pre-training and meta-learning phases are not yet well-understood, and making progress on understanding this will shed light on the most promising directions for few-shot classification.

Also, using the Soft-Nearest-Neighbor-Loss is an interesting property to measure (and to try and reinforce) in the pre-trained representations.

Cons
====
[A] The biggest weakness of this work, in my opinion, is the lack of connection with previous work that is very similar. Specifically, the property that is referred to in this paper as ‘disentanglement’ is very related to previous notions that have been studied in this context of pre-training for few-shot learning. Specifically, [2] argued that the success of the cosine classifier during pre-training (compared to a standard classifier) is due to explicitly minimizing the intra-class variance of each class (which leads to better clustering, and to better “‘disentanglement” in the way that the term is used in this paper).

Pushing that direction further, [1] proposed regularizers whose purpose is to directly encourage the pre-training phase to have this property of better clustering: minimizing the intra-class variance and maximizing the inter-class variance. This paper should be discussed as related work and should be compared to experimentally since their approach is very similar to the one proposed here.

[B] Unfortunately I also found the writing to be of poor quality. Some minor grammatical or wording errors did not distract me too much from understanding the intended meaning, but there were certain statements which I found hard to understand, or disagreed with. Some examples are below:

“Due to episodic training, meta-learning methods generalize better than traditional transfer-like methods for the novel classes”. The jury is actually still out on this, so I don’t think it’s appropriate to make this claim. Better generalization was the motivation of episodic models, indeed, but in practice non-episodic approaches have been shown to perform quite well, as pointed out in this paper too.

“Episodic sampling is time-consuming”. Can you explain why that is? I don’t disagree (based on my experience too) but I don’t believe it is obvious, and it would be useful to explain this.

“In the previous section, we conclude that the last layer in the backbone would be more disentangled after episodic training”. It’s unclear to me how that conclusion follows.

[C] Another weakness is that the proposed method does not perform too strongly compared to the baselines / previous methods. It seems that the gain is larger for smaller architectures, which is in line with the observation in [2] that minimizing intra-task variance is most beneficial for small backbones. On the other hand, for larger architectures, RP-Proto is not better than plain proto (notice the overlap in the confidence intervals in the respective entries of Table 1).

[D] I also found the experiments to be weak in terms of the analysis of disentanglement during pre-training and episodic training. Contrary to the authors’ observation, it doesn’t really seem to me that the disentanglement loss is going down too much during episodic training by looking at Figure 1. The conv and resnet18 curves are mostly flat. The resnet10 one does go down noticeably but then starts going up again. I’m also not sure what causes the large discrepancy in the behavior of the resnet10 curve compared to the other two? My initial thought was network capacity, but resnet 10’s capacity is in between that of the other two networks if I understand correctly, so it’s hard to draw a conclusion there.

Overall
======
I vote for rejection of this paper in its current form, mostly due to the missing comparison with the very related method mentioned above, the quality of the writing and the weakness of the experimental results, as described above.

Suggestion for additional experiments
============================
Table 1 shows that MetaOptNet outperforms RP-Proto (I’m looking at the entry of RP-Proto with the ResNet12 backbone for an apples-to-apples comparison with MetaOptNet). I would be curious to see an RP-MetaOptNet variant too. More generally, does the proposed regularizer also lead to improvements in episodic approaches that are closer to state-of-the-art compared to Prototypical Networks?

Further, as an additional data point, it would be useful to also report the performance of the pre-trained network itself on the few-shot test tasks (without a meta-learning phase at all). For an apples-to-apples comparison with the reported Prototypical Network variants, Prototypical Networks can be used to solve each test task still, but operating directly on top of the representation learned from pre-training, instead of the representation produced by the episodic phase.

Additional comments for fixing minor issues and improving clarity
===============================================
Below are a few more recommendations and singled-out sentences that I think should be re-written to improve clarity.

In the “Mixed Framework” section, [3] should also be cited among the papers that used a pre-trained solution as the initialization for the meta-learning stage as this is how the meta-learners in that paper were trained as well.

The provided reference for fo-MAML is incorrect. fo-MAML was actually introduced in the original MAML paper (Finn et al, 2017). The provided (Nichol et al., 2018) reference introduced Reptile, which is similar but not the same as fo-MAML.

“Episodic training is key to make meta-learning prominent”. I’m not sure what this sentence means. Are there other ways of meta-learning in this context without episodic training?

The description of optimization-based meta-learning: “[...] try to get an embedding that could easily fit subtasks by adding some extra layers”. This is not entirely accurate. MAML, for instance, does not add any extra layers per task. Instead, the entire network is rapidly fine-tuned within each task as well as meta-learned across tasks.

References
=========
[1] Unraveling Meta-Learning: Understanding Feature Representations for Few-Shot Tasks. Goldblum et al. ICML 2020.
[2] A Closer Look at Few-shot Classification. Chen et al. ICLR 2019.
[3] Meta-Dataset: A Dataset of Datasets for Learning to Learn from Few Examples. Triantafillou et al. ICLR 2020.

---

### Official Review · AnonReviewer3 · 2020-10-28
**On the Role of Pre-training for Meta Few-Shot Learning**

**Rating:** 5
**Confidence:** 5

**Review:**

Paper summary

In this paper the authors provide a summary of the role of pre-training in meta-learning. They investigate the performance implications of pretraining for a common pretraining method, prototypical networks, and propose an additional regularization loss to improve the generalization of pre-training. They evaluate the proposed method on miniimagenet and cifar100

---------------------------------------------------------------------------------------------------------------------------
Positives and negatives

+ The summary of the pretraining role in meta-learning is accurate, and the proposed idea of using soft-nearest neighbor as a regularization term in pretraining is novel.
+ The method is simple to understand and implement.
+ The experiments section is well documented.
+ I like the algorithm explanation of the method.
- The paper should mention that pre-training in meta-learning is not at all a novel idea, many papers from couple of years ago use pretraining to initialize the weights of their convolutional backbone (see for example LEO, which is cited by the authors). Therefore the authors should make it clear that the only novel idea here is the use of Soft nearest neighbor loss as a regularization.
Given the above and if we look at table 1, we can see that the soft nearest neighbor loss does not actually improve the accuracy compared to simple pretraining for larger backbones. Therefore I would consider the significance of this result very small, and therefore further study should be done to see if it helps under different conditions (e.g. different datasets, more limited data regimes, etc).
- Given the small size of the datasets under consideration, I am surprised the experiments section is so weak. I would expect to see experiments for more datasets and more meta-learning methods, e.g. maml, or matching networks, or even simple linear fit on the embeddings). For an example of a more thorough set of experiments, refer to https://arxiv.org/abs/1910.01319, which also looks at pretraining in the context of meta-learning but examines many more different backbone architectures, meta-learning algorithms, and datasets.

---------------------------------------------------------------------------------------------------------------------------
Recommendation

As it stands this paper requires a significant rewrite to meet the standards of this conference. Therefore I recommend this paper be rejected. The following aspects should be improved: the paper should be clear that the novel contribution is the soft-nearest neighbor regularization, and therefore the major focus of the experiments section should be on ablating this loss and testing it extensively under different conditions (e.g. more datasets, different meta-learning methods, low data regime, etc). Under the tested conditions, the proposed soft-nearest neighbor regularization does not seem to help much. The paper also should be thoroughly reviewed and rewritten as it was very hard to follow and had several sentences which are not possible to understand (a few examples below).

---------------------------------------------------------------------------------------------------------------------------
Questions

* Figure 1, what is the scale of SNN loss? From the y-scale it looks like it varies from 11.5 to 11.44. I have no idea whether this is basically just noise. Would be good to explain either in the caption or figure what is an SNN value for fully disentangled and fully entangled data.
* I didn’t understand what the authors mean by the “parallelism property”. Section 2.4. Could you elaborate further on it? The episodes in episodic training are also trained in a batched SDG setting so it would consider them “parallel” in a sense.
* In the last sentence of 2.4 the authors say “for shallow and deep backbones, it increases performance”, however from table 1 I read that it only increases performance for shallow backbones. Could the authors justify this sentence?
---------------------------------------------------------------------------------------------------------------------------
Feedback (not related to the score)

* Table 1, would be good to have an explanation of what is the hyperparameter C. The only mention of it is in Eq. 5 as the exponent of alpha, so it can be understood as the regularization but it would be nice to explain in both in the main text under eq. 5 as well as in the caption for Table 1.
* In general I would recommend the authors write more in the figure captions. For example figure 3 and figure 4 are very hard to understand, and a few sentences explaining the point of the figure would really help the reader.
* To improve the strength of the paper, I would suggest focusing on the disentanglement properties of the SNN regularization and, as proposed above, explore it under a lot more settings. In addition, I would look at how this SNN regularization affects domain transfer as I suspect it could actually be more useful in that case (e.g. pretrain on miniimagenet, test on CIFAR or even MNIST)

---

### Official Review · AnonReviewer4 · 2020-10-29

**Rating:** 4
**Confidence:** 4

**Review:**

#### Summary

The submission attempts to understand the role of episodic fine-tuning in a few-shot classification context.

Using Prototypical Networks as a case study, the authors measure the entanglement of class representations (using a soft nearest neighbour measure as was done by Frosst et al. (2019) in the supervised learning case) during episodic fine-tuning for several backbone architectures fine-tuned on 5-way 5-shot mini-ImageNet episodes. Based on these observations, the paper concludes that episodic fine-tuning tends to decrease entanglement in the penultimate layer and proposes to add a regularization term to the supervised pre-training phase which encourages disentanglement in that layer.

Results are presented on the mini-ImageNet benchmark and the proposed approach is claimed to perform on-par with or better than competing approaches. Accuracy curves are also shown to support the claim that the proposed approach requires training on less episodes.

#### Strengths and weaknesses

* **+** The paper’s topic is very relevant to questions being raised in the recent literature regarding the differences between supervised and episodic training.
* **+** The premise of studying the role of episodic fine-tuning from the perspective of feature (dis)entanglement is an interesting application of Frosst et al. (2019)’s work to the few-shot classification setting.
* **-** Vague or inconsistent use of terminology.
* **-** Poor writing and presentation.
* **-** Performance of the proposed approach is not competitive with competing approaches, contrary to what's claimed in the submission.

#### Recommendation

I recommend rejection. While the paper’s premise is interesting, the submission suffers from poor writing and presentation quality, and I’m not entirely convinced by the claimed causal relationship between representation (dis)entanglement and episodic fine-tuning.

#### Detailed justification

My main concerns have to do with the presentation of the results and the interpretation of episodic fine-tuning as a way to decrease the representation’s entanglement.

The list of competing approaches in Table 1 is incomplete and outdated. For instance, Tian et al. (2020)’s "Rethinking Few-Shot Image Classification: a Good Embedding Is All You Need?" obtains around 64.8% on mini-ImageNet 5-way 1-shot using a ResNet-12 architecture. Tian et al. (2020) also lists several approaches using a Conv4 backbone that achieve a mini-ImageNet 5-way 1-shot performance greater than 50.4%. I therefore disagree with the assertion that "our method shows competitive performance to other methods" and that "when the backbone is shallow, we have outperformed all other methods with the same backbone".

The paper hints at the fact that the Euclidean metric used by Prototypical Networks doesn’t work well with highly entangled representations. Another possible interpretation is that the backbone is pre-trained using a linear output layer, which computes the inner-product between the representation and class weight vectors, and that the squared distance computed by the Euclidean metric is not well suited to the resulting representation. According to that interpretation, part of what episodic fine-tuning does is correct for this mismatch in metrics between (meta-)training and (meta-)testing. I think the paper’s interpretation would be more convincing if the pre-trained backbone used a quadratic output layer (i.e. the logits are computed as the negative squared distance between the representations and class weight vectors) and therefore controlled for metric mismatch.

I have additional issues with the way in which results are presented:

* Section 3.3 mentions a result presented in Section 4.4, then Section 3.4 begins by stating that the previous section concludes that the last layer in the backbone is more disentangled after episodic training. This means that in order to have the proper context to understand Section 3.4, the reader needs to jump forward and read Section 4.4. I recommend changing the order of the presentation so that it is more linear.
* In Table 1, 95% confidence intervals are provided, but the absence of identification of the best-performing approach(es) in each setting makes it hard to draw high-level conclusions at a glance. I would suggest bolding the best accuracy in each column along with all other entries for which a 95% confidence interval test on the difference between the means is inconclusive in determining that the difference is significant.

Overall, the submission could benefit from another round of careful proofreading. It contains several grammar mistakes which, while they do not significantly compromise clarity, make reading the paper harder than it should be. Examples include:

* "[...] and crafted the hard episode to make necessary episodes fewer."
* "Disentanglement is the property whether the data-points [...]"
* "Benefited by the understanding, [...]"

I also noted a few false or unsupported statements:

* "Due to episodic training, meta-learning methods generalize better than traditional transfer-like methods for the novel classes." Can the authors expand on this? What do they mean by "generalize"? Do they refer to test classes from the same domain (e.g. mini-ImageNet test classes), or test classes from different domains (i.e. cross-domain generalization)? I don’t see the statement as generally accepted, especially given the many recent papers that show strong performance with well-tuned transfer learning baselines.
* "Soft-Nearest Neighbour loss [...] is proposed by Frosst et al. (2019)" Frosst et al. credits Salakhutdinov and Hinton (2007) for the soft nearest neighbour loss, which itself draws inspiration from Goldberger et al. (2005)’s Neighbourhood Component Analysis.
* "Though the split is quite naive, the afterward episodic learning shows promising improvement." Can the authors point to work that provides empirical evidence for this statement?

Finally, the submission’s use of terminology is vague or inconsistent at times:

* The categorization of approaches as either "supervised pre-training" or "meta few-shot learning" feels incomplete to me. While some approaches (most recently Meta-Baseline and Meta-Dataset’s few-shot learners) do perform supervised pre-training followed by episodic fine-tuning, most well-known approaches such as Matching Networks, Prototypical Networks, MAML, etc. do not prescribe a supervised pre-training phase.
* The term "meta few-shot learning" is not widely used in the literature and appears to be introduced in this paper as far as I can tell. It’s defined in the introduction to be the combination of supervised pre-training and episodic fine-tuning, but it’s also used in Section 3.2 to categorize Prototypical Networks, whose formulation does *not* prescribe a supervised pre-training phase.

#### Questions

1. What do the authors mean when they state that "[due] to the parallelism property, normal training literature is much faster than episodic training"? Isn’t the forward propagation through the embedding function in a few-shot learner such as Prototypical Networks just as parallelizable as the forward propagation in a supervised classifier?
1. The submission claims that the proposed approach alleviates the burden of training episodically on a large number of episodes. Given the performance of well-tuned supervised baselines, why should we perform episodic fine-tuning? Shouldn’t it be sufficient to use the pre-trained backbone as-is?
1. The submission follows Chen et al. (2020) for the pre-training and episodic fine-tuning procedure but doesn’t compare against it in Table 1. How come?

#### Additional feedback

1. This is arguably inconsequential, but the paper motivates few-shot learning using bird classification as an example, stating that "an ornithologist typically can only obtain a few pictures per bird species". I don’t know if I agree that this is typically the case: looking at the iNaturalist online database, thousands of pictures can be found for a large number of bird species.
1. The related work section appears to contain mostly pre-2020 references and does not mention recent work such as (Simple) CNAPs, SUR, CrossTransformers, etc.

---

### Official Review · AnonReviewer2 · 2020-10-30
**Neat method, a few questions**

**Rating:** 7
**Confidence:** 3

**Review:**

This paper studies how a regularization loss that measures the distance between feature $x$ and learnable class embedding $W_y$ helps meta-learning. The authors show that the SNN loss is lower when the regularization loss is in use in the meta-training phase, which supports the authors' conjecture that this regularization helps disentanglement of the penultimate layer in the backbone network, thus enhancing episodic learning.

Pros:
1. The method is very neat and easy to implement
2. The experiment results are mostly in favor of the authors' claims

Questions:

1. Not sure if the term "pre-training" is proper here. The main focus of this paper, is the regularization term $l_{reg}$. Why don't tell the audience about this in a more straightforward way?
2. Should it be $N$-way $K$-shot, why $1 \leq y_i\leq \mathbf{K}$? I believe there's a typo.
3. In Figure 1, all of the three lines are of some sort of V-shaped, i.e. they grew up a bit after reaching minima in the middle. Do you think might be the reason for this phenomenon?

---

### Decision · Program_Chairs · 2021-01-07
**Final Decision**

**Decision:**

Reject

**Comment:**

There is value in analyzing pre-training for few-shot learning, and the observation that improved disentanglement might lead to better initialization schemes for few-shot learners is worth exploring. However, in its current state, the reviewers do not think the paper is ready for publication. Specifically, work needs to be done to improve the clarity, comparison to related work, and experimental analysis.